# `RETAIN`: An Interpretable Predictive Model for Healthcare using Reverse Time Attention Mechanism

**Edward Choi**[*], **Mohammad Taha Bahadori**[*], **Joshua A. Kulas**[*],
**Andy Schuetz**[†], **Walter F. Stewart**[†], **Jimeng Sun**[*]
[*] Georgia Institute of Technology      [†] Sutter Health
{mp2893,bahadori,jkulas3}@gatech.edu,
{schueta1,stewarwf}@sutterhealth.org, jsun@cc.gatech.edu

## Abstract

Accuracy and interpretability are two dominant features of successful predictive models. Typically, a choice must be made in favor of complex black box models such as recurrent neural networks (RNN) for accuracy versus less accurate but more interpretable traditional models such as logistic regression. This tradeoff poses challenges in medicine where both accuracy and interpretability are important. We addressed this challenge by developing the REverse Time AttentIoN model (`RETAIN`) for application to Electronic Health Records (EHR) data. `RETAIN` achieves high accuracy while remaining clinically interpretable and is based on a two-level neural attention model that detects influential past visits and significant clinical variables within those visits (e.g. key diagnoses). `RETAIN` mimics physician practice by attending the EHR data in a reverse time order so that recent clinical visits are likely to receive higher attention. `RETAIN` was tested on a large health system EHR dataset with 14 million visits completed by 263K patients over an 8 year period and demonstrated predictive accuracy and computational scalability comparable to state-of-the-art methods such as RNN, and ease of interpretability comparable to traditional models.

## 1 Introduction

The broad adoption of Electronic Health Record (EHR) systems has opened the possibility of applying clinical predictive models to improve the quality of clinical care. Several systematic reviews have underlined the care quality improvement using predictive analysis [7, 25, 5, 20]. EHR data can be represented as temporal sequences of high-dimensional clinical variables (e.g., diagnoses, medications and procedures), where the sequence ensemble represents the documented content of medical visits from a single patient. Traditional machine learning tools summarize this ensemble into aggregate features, ignoring the temporal and sequence relationships among the feature elements. The opportunity to improve both predictive accuracy and interpretability is likely to derive from effectively modeling temporality and high-dimensionality of these event sequences.

Accuracy and interpretability are two dominant features of successful predictive models. There is a common belief that one has to trade accuracy for interpretability using one of three types of traditional models [6]: 1) identifying a set of rules (*e.g.* via decision trees [27]), 2) case-based reasoning by finding similar patients (*e.g.* via $k$-nearest neighbors [18] and distance metric learning [36]), and 3) identifying a list of risk factors (*e.g.* via LASSO coefficients [15]). While interpretable, all of these models rely on aggregated features, ignoring the temporal relation among features inherent to EHR data. As a consequence, model accuracy is sub-optimal. Latent-variable time-series models, such as [34, 35], account for temporality, but often have limited interpretation due to abstract state variables.

Recently, recurrent neural networks (RNN) have been successfully applied in modeling sequential EHR data to predict diagnoses [30] and disease progression [11, 14]. But, the gain in accuracy from

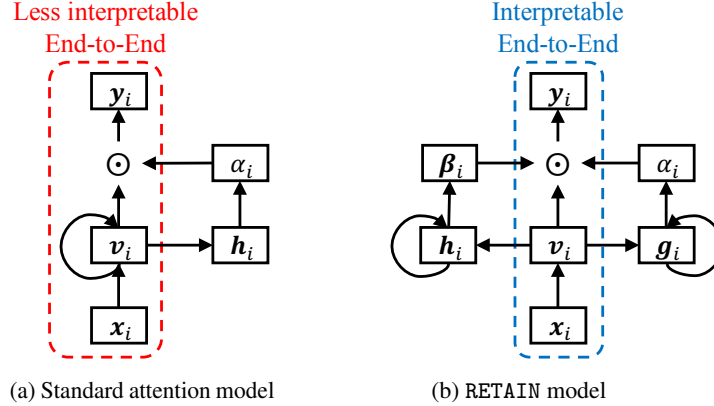

Figure 1: Common attention models vs. RETAIN, using folded diagrams of RNNs. (a) Standard attention mechanism: the recurrence on the hidden state vector $\mathbf{v}_i$ hinders interpretation of the model. (b) Attention mechanism in RETAIN: The recurrence is on the attention generation components ($\mathbf{h}_i$ or $\mathbf{g}_i$) while the hidden state $\mathbf{v}_i$ is generated by a simpler more interpretable output.

use of RNNs is at the cost of model output that is notoriously difficult to interpret. While there have been several attempts at directly interpreting RNNs [19, 26, 8], these methods are not sufficiently developed for application in clinical care.

We have addressed this limitation using a modeling strategy known as RETAIN, a two-level neural attention model for sequential data that provides detailed interpretation of the prediction results while retaining the prediction accuracy comparable to RNN. To this end, RETAIN relies on an attention mechanism modeled to represent the behavior of physicians during an encounter. A distinguishing feature of RETAIN (see Figure 1) is to leverage sequence information using an attention generation mechanism, while learning an interpretable representation. And emulating physician behaviors, RETAIN examines a patient's past visits in reverse time order, facilitating a more stable attention generation. As a result, RETAIN identifies the most meaningful visits and quantifies visit specific features that contribute to the prediction.

RETAIN was tested on a large health system EHR dataset with 14 million visits completed by 263K patients over an 8 year period. We compared predictive accuracy of RETAIN to traditional machine learning methods and to RNN variants using a case-control dataset to predict a future diagnosis of heart failure. The comparative analysis demonstrates that RETAIN achieves comparable performance to RNN in both accuracy and speed and significantly outperforms traditional models. Moreover, using a concrete case study and visualization method, we demonstrate how RETAIN offers an intuitive interpretation.

## 2   Methodology

We first describe the structure of sequential EHR data and our notation, then follow with a general framework for predictive analysis in healthcare using EHR, followed by details of the RETAIN method.

**EHR Structure and our Notation.** The EHR data of each patient can be represented as a time-labeled sequence of multivariate observations. Assuming we use $r$ different variables, the $n$-th patient of $N$ total patients can be represented by a sequence of $T^{(n)}$ tuples $(t_i^{(n)}, \mathbf{x}_i^{(n)}) \in \mathbb{R} \times \mathbb{R}^r, i = 1, \ldots, T^{(n)}$. The timestamps $t_i^{(n)}$ denotes the time of the $i$-th visit of the $n$-th patient and $T^{(n)}$ the number of visits of the $n$-th patient. To minimize clutter, we describe the algorithms for a single patient and have dropped the superscript $(n)$ whenever it is unambiguous. The goal of predictive modeling is to predict the label at each time step $\mathbf{y}_i \in \{0, 1\}^s$ or at the end of the sequence $\mathbf{y} \in \{0, 1\}^s$. The number of labels $s$ can be more than one.

For example, in disease progression modeling (DPM) [11], each visit of a patient visit sequence is represented by a set of varying number of medical codes $\{c_1, c_2, \ldots, c_n\}$. $c_j$ is the $j$-th code from the vocabulary $\mathcal{C}$. Therefore, in DPM, the number of variables $r = |\mathcal{C}|$ and input $\mathbf{x}_i \in \{0, 1\}^{|\mathcal{C}|}$ is

a binary vector where the value one in the $j$-th coordinate indicates that $c_j$ was documented in $i$-th visit. Given a sequence of visits $\mathbf{x}_1, \ldots, \mathbf{x}_T$, the goal of DPM is, for each time step $i$, to predict the codes occurring at the next visit $\mathbf{x}_2, \ldots, \mathbf{x}_{T+1}$, with the number of labels $s = |\mathcal{C}|$.

In case of learning to diagnose (L2D) [30], the input vector $\mathbf{x}_i$ consists of continuous clinical measures. If there are $r$ different measurements, then $\mathbf{x}_i \in \mathbb{R}^r$. The goal of L2D is, given an input sequence $\mathbf{x}_1, \ldots, \mathbf{x}_T$, to predict the occurrence of a specific disease ($s = 1$) or multiple diseases ($s > 1$). Without loss of generality, we will describe the algorithm for DPM, as L2D can be seen as a special case of DPM where we make a single prediction at the end of the visit sequence.

In the rest of this section, we will use the abstract symbol RNN to denote any recurrent neural network variants that can cope with the vanishing gradient problem [3], such as LSTM [23], GRU [9], and IRNN [29], with any depth (number of hidden layers).

## 2.1 Preliminaries on Neural Attention Models

Attention based neural network models are being successfully applied to image processing [1, 32, 21, 37], natural language processing [2, 22, 33] and speech recognition [12]. The utility of the attention mechanism can be seen in the language translation task [2] where it is inefficient to represent an entire sentence with one fixed-size vector because neural translation machines finds it difficult to translate the given sentence represented by a single vector.

Intuitively, the attention mechanism for language translation works as follows: given a sentence of length $S$ in the original language, we generate $\mathbf{h}_1, \ldots, \mathbf{h}_S$, to represent the words in the sentence. To find the $j$-th word in the target language, we generate attentions $\alpha_i^j$ for $i = 1, \ldots, S$ for each word in the original sentence. Then, we compute the context vector $\mathbf{c}_j = \sum_i \alpha_i^j \mathbf{h}_i$ and use it to predict the $j$-th word in the target language. In general, the attention mechanism allows the model to focus on a specific word (or words) in the given sentence when generating each word in the target language.

We rely on a conceptually similar temporal attention mechanism to generate interpretable prediction models using EHR data. Our model framework is motivated by and mimics how doctors attend to a patient's needs and explore the patient record, where there is a focus on specific clinical information (e.g., key risk factors) working from the present to the past.

## 2.2 Reverse Time Attention Model RETAIN

Figure 2 shows the high-level overview of our model, where a central feature is to delegate a considerable portion of the prediction responsibility to the process for generating attention weights. This is intended to address, in part, the difficulty with interpreting RNNs where the recurrent weights feed past information to the hidden layer. Therefore, to consider both the visit-level and the variable-level (individual coordinates of $\mathbf{x}_i$) influence, we use a linear embedding of the input vector $\mathbf{x}_i$. That is, we define

$$\mathbf{v}_i = \mathbf{W}_{emb}\mathbf{x}_i, \qquad \text{(Step 1)}$$

where $\mathbf{v}_i \in \mathbb{R}^m$ denotes the embedding of the input vector $\mathbf{x}_i \in \mathbb{R}^r$, $m$ the size of the embedding dimension, $\mathbf{W}_{emb} \in \mathbb{R}^{m \times r}$ the embedding matrix to learn. We can alternatively use more sophisticated yet interpretable representations such as those derived from multilayer perceptron (MLP) [13, 28]. MLP has been used for representation learning in EHR data [10].

We use two sets of weights, one for the visit-level attention and the other for variable-level attention, respectively. The scalars $\alpha_1, \ldots, \alpha_i$ are the visit-level attention weights that govern the influence of each visit embedding $\mathbf{v}_1, \ldots, \mathbf{v}_i$. The vectors $\boldsymbol{\beta}_1, \ldots, \boldsymbol{\beta}_i$ are the variable-level attention weights that focus on each coordinate of the visit embedding $v_{1,1}, v_{1,2}, \ldots, v_{1,m}, \ldots, v_{i,1}, v_{i,2}, \ldots, v_{i,m}$.

We use two RNNs, $\text{RNN}_\alpha$ and $\text{RNN}_\beta$, to separately generate $\alpha$'s and $\boldsymbol{\beta}$'s as follows,

$$\mathbf{g}_i, \mathbf{g}_{i-1}, \ldots, \mathbf{g}_1 = \text{RNN}_\alpha(\mathbf{v}_i, \mathbf{v}_{i-1}, \ldots, \mathbf{v}_1),$$
$$e_j = \mathbf{w}_\alpha^\top \mathbf{g}_j + b_\alpha, \quad \text{for} \quad j = 1, \ldots, i$$
$$\alpha_1, \alpha_2, \ldots, \alpha_i = \text{Softmax}(e_1, e_2, \ldots, e_i) \qquad \text{(Step 2)}$$
$$\mathbf{h}_i, \mathbf{h}_{i-1}, \ldots, \mathbf{h}_1 = \text{RNN}_{\boldsymbol{\beta}}(\mathbf{v}_i, \mathbf{v}_{i-1}, \ldots, \mathbf{v}_1)$$
$$\boldsymbol{\beta}_j = \tanh\left(\mathbf{W}_{\boldsymbol{\beta}}\mathbf{h}_j + \mathbf{b}_{\boldsymbol{\beta}}\right) \quad \text{for} \quad j = 1, \ldots, i, \qquad \text{(Step 3)}$$

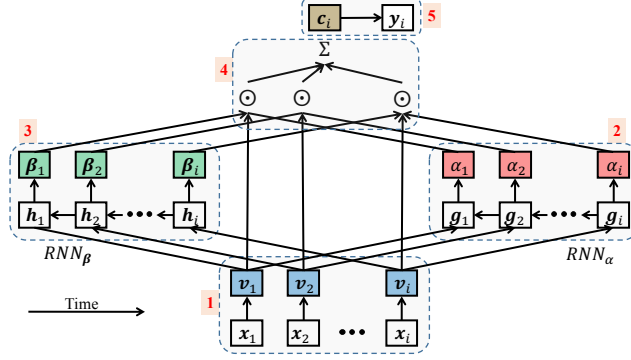

Figure 2: Unfolded view of `RETAIN`'s architecture: Given input sequence $\mathbf{x}_1, \ldots, \mathbf{x}_i$, we predict the label $\mathbf{y}_i$. **Step 1**: Embedding, **Step 2**: generating $\alpha$ values using $\text{RNN}_\alpha$, **Step 3**: generating $\boldsymbol{\beta}$ values using $\text{RNN}_\beta$, **Step 4**: Generating the context vector using attention and representation vectors, and **Step 5**: Making prediction. Note that in Steps 2 and 3 we use RNN in the reversed time.

where $\mathbf{g}_i \in \mathbb{R}^p$ is the hidden layer of $\text{RNN}_\alpha$ at time step $i$, $\mathbf{h}_i \in \mathbb{R}^q$ the hidden layer of $\text{RNN}_\beta$ at time step $i$ and $\mathbf{w}_\alpha \in \mathbb{R}^p, b_\alpha \in \mathbb{R}, \mathbf{W}_\beta \in \mathbb{R}^{m \times q}$ and $\mathbf{b}_\beta \in \mathbb{R}^m$ are the parameters to learn. The hyperparameters $p$ and $q$ determine the hidden layer size of $\text{RNN}_\alpha$ and $\text{RNN}_\beta$, respectively. Note that for prediction at each timestamp, we generate a new set of attention vectors $\alpha$ and $\boldsymbol{\beta}$. For simplicity of notation, we do not include the index for predicting at different time steps. In Step 2, we can use Sparsemax [31] instead of Softmax for sparser attention weights.

As noted, `RETAIN` generates the attention vectors by running the RNNs backward in time; i.e., $\text{RNN}_\alpha$ and $\text{RNN}_\beta$ both take the visit embeddings in a reverse order $\mathbf{v}_i, \mathbf{v}_{i-1}, \ldots, \mathbf{v}_1$. Running the RNN in reversed time order also offers computational advantages since the reverse time order allows us to generate $e$'s and $\boldsymbol{\beta}$'s that dynamically change their values when making predictions at different time steps $i = 1, 2, \ldots, T$. This ensures that the attention vectors are modified at each time step, increasing the computational stability of the attention generation process.[1]

Using the generated attentions, we obtain the context vector $\mathbf{c}_i$ for a patient up to the $i$-th visit as follows,

$$\mathbf{c}_i = \sum_{j=1}^{i} \alpha_j \boldsymbol{\beta}_j \odot \mathbf{v}_j, \qquad \text{(Step 4)}$$

where $\odot$ denotes element-wise multiplication. We use the context vector $\mathbf{c}_i \in \mathbb{R}^m$ to predict the true label $\mathbf{y}_i \in \{0, 1\}^s$ as follows,

$$\widehat{\mathbf{y}}_i = \text{Softmax}(\mathbf{W}\mathbf{c}_i + \mathbf{b}), \qquad \text{(Step 5)}$$

where $\mathbf{W} \in \mathbb{R}^{s \times m}$ and $\mathbf{b} \in \mathbb{R}^s$ are parameters to learn. We use the cross-entropy to calculate the classification loss as follows,

$$\mathcal{L}(\mathbf{x}_1, \ldots, \mathbf{x}_T) = -\frac{1}{N} \sum_{n=1}^{N} \frac{1}{T^{(n)}} \sum_{i=1}^{T^{(n)}} \left( \mathbf{y}_i^\top \log(\widehat{\mathbf{y}}_i) + (\mathbf{1} - \mathbf{y}_i)^\top \log(\mathbf{1} - \widehat{\mathbf{y}}_i) \right) \qquad (1)$$

where we sum the cross entropy errors from all dimensions of $\widehat{\mathbf{y}}_i$. In case of real-valued output $\mathbf{y}_i \in \mathbb{R}^s$, we can change the cross-entropy in Eq. (1) to, for example, mean squared error.

Overall, our attention mechanism can be viewed as the inverted architecture of the standard attention mechanism for NLP [2] where the words are encoded by RNN and the attention weights are generated by MLP. In contrast, our method uses MLP to embed the visit information to preserve interpretability and uses RNN to generate two sets of attention weights, recovering the sequential information as well as mimicking the behavior of physicians. Note that we did not use the timestamp of each visit in our formulation. Using timestamps, however, provides a small improvement in the prediction performance. We propose a method to use timestamps in Appendix A.

## 3    Interpreting `RETAIN`

Finding the visits that contribute to prediction are derived using the largest $\alpha_i$, which is straightforward. However, finding influential variables is slightly more involved as a visit is represented by an ensemble of medical variables, each of which can vary in its predictive contribution. The contribution of each variable is determined by $\mathbf{v}$, $\boldsymbol{\beta}$ and $\alpha$, and interpretation of $\alpha$ alone informs which visit is influential in prediction but not why.

We propose a method to interpret the end-to-end behavior of `RETAIN`. By keeping $\alpha$ and $\boldsymbol{\beta}$ values fixed as the attention of doctors, we analyze changes in the probability of each label $y_{i,1}, \ldots, y_{i,s}$ in relation to changes in the original input $x_{1,1}, \ldots, x_{1,r}, \ldots, x_{i,1}, \ldots, x_{i,r}$. The $x_{j,k}$ that yields the largest change in $y_{i,d}$ will be the input variable with highest contribution. More formally, given the sequence $\mathbf{x}_1, \ldots, \mathbf{x}_i$, we are trying to predict the probability of the output vector $\mathbf{y}_i \in \{0,1\}^s$, which can be expressed as follows

$$p(\mathbf{y}_i|\mathbf{x}_1, \ldots, \mathbf{x}_i) = p(\mathbf{y}_i|\mathbf{c}_i) = \text{Softmax}\left(\mathbf{W}\mathbf{c}_i + \mathbf{b}\right) \tag{2}$$

where $\mathbf{c}_i \in \mathbb{R}^m$ denotes the context vector. According to Step 4, $\mathbf{c}_i$ is the sum of the visit embeddings $\mathbf{v}_1, \ldots, \mathbf{v}_i$ weighted by the attentions $\alpha$'s and $\boldsymbol{\beta}$'s. Therefore Eq (2) can be rewritten as follows,

$$p(\mathbf{y}_i|\mathbf{x}_1, \ldots, \mathbf{x}_i) = p(\mathbf{y}_i|\mathbf{c}_i) = \text{Softmax}\left(\mathbf{W}\Big(\sum_{j=1}^{i} \alpha_j \boldsymbol{\beta}_j \odot \mathbf{v}_j\Big) + \mathbf{b}\right) \tag{3}$$

Using the fact that the visit embedding $\mathbf{v}_i$ is the sum of the columns of $\mathbf{W}_{emb}$ weighted by each element of $\mathbf{x}_i$, Eq (3) can be rewritten as follows,

$$p(\mathbf{y}_i|\mathbf{x}_1, \ldots, \mathbf{x}_i) = \text{Softmax}\left(\mathbf{W}\Big(\sum_{j=1}^{i} \alpha_j \boldsymbol{\beta}_j \odot \sum_{k=1}^{r} x_{j,k}\mathbf{W}_{emb}[:,k]\Big) + \mathbf{b}\right)$$

$$= \text{Softmax}\left(\sum_{j=1}^{i}\sum_{k=1}^{r} x_{j,k}\, \alpha_j \mathbf{W}\Big(\boldsymbol{\beta}_j \odot \mathbf{W}_{emb}[:,k]\Big) + \mathbf{b}\right) \tag{4}$$

where $x_{j,k}$ is the $k$-th element of the input vector $\mathbf{x}_j$. Eq (4) can be completely deconstructed to the variables at each input $\mathbf{x}_1, \ldots, \mathbf{x}_i$, which allows for calculating the contribution $\omega$ of the $k$-th variable of the input $\mathbf{x}_j$ at time step $j \leq i$, for predicting $\mathbf{y}_i$ as follows,

$$\omega(\mathbf{y}_i, x_{j,k}) = \underbrace{\alpha_j \mathbf{W}(\boldsymbol{\beta}_j \odot \mathbf{W}_{emb}[:,k])}_{\text{Contribution coefficient}} \underbrace{x_{j,k}}_{\text{Input value}}, \tag{5}$$

where the index $i$ of $\mathbf{y}_i$ is omitted in the $\alpha_j$ and $\boldsymbol{\beta}_j$. As we have described in Section 2.2, we are generating $\alpha$'s and $\boldsymbol{\beta}$'s at time step $i$ in the visit sequence $\mathbf{x}_1, \ldots, \mathbf{x}_T$. Therefore the index $i$ is always assumed for $\alpha$'s and $\boldsymbol{\beta}$'s. Additionally, Eq (5) shows that when we are using a binary input value, the coefficient itself is the contribution. However, when we are using a non-binary input value, we need to multiply the coefficient and the input value $x_{j,k}$ to correctly calculate the contribution.

## 4    Experiments

We compared performance of `RETAIN` to RNNs and traditional machine learning methods. Given space constraints, we only report the results on the learning to diagnose (L2D) task and summarize the disease progression modeling (DPM) in Appendix C. The `RETAIN` source code is publicly available at https://github.com/mp2893/retain.

### 4.1    Experimental setting

**Source of data:** The dataset consists of electronic health records from Sutter Health. The patients are 50 to 80 years old adults chosen for a heart failure prediction model study. From the encounter records, medication orders, procedure orders and problem lists, we extracted visit records consisting of diagnosis, medication and procedure codes. To reduce the dimensionality while preserving the clinical information, we used existing medical groupers to aggregate the codes into input variables. The details of the medical groupers are given in the Appendix B. A profile of the dataset is summarized in Table 1.

Table 1: Statistics of EHR dataset. (D:Diagnosis, R:Medication, P:Procedure)

| # of patients | 263,683 | Avg. # of codes in a visit | 3.03 |
|---|---|---|---|
| # of visits | 14,366,030 | Max # of codes in a visit | 62 |
| Avg. # of visits per patient | 54.48 | Avg. # of Dx codes in a visit | 1.83 |
| # of medical code groups | 615 (D:283, R:94, P:238) | Max # of Dx in a visit | 42 |

**Implementation details:** We implemented `RETAIN` with Theano 0.8 [4]. For training the model, we used Adadelta [38] with the mini-batch of 100 patients. The training was done in a machine equipped with Intel Xeon E5-2630, 256GB RAM, two Nvidia Tesla K80's and CUDA 7.5.

**Baselines:** For comparison, we completed the following models.

- **Logistic regression (LR)**: We compute the counts of medical codes for each patient based on all her visits as input variables and normalize the vector to zero mean and unit variance. We use the resulting vector to train the logistic regression.
- **MLP**: We use the same feature construction as **LR**, but put a hidden layer of size 256 between the input and output.
- **RNN**: RNN with two hidden layers of size 256 implemented by the GRU. Input sequences $\mathbf{x}_1, \ldots, \mathbf{x}_i$ are used. Logistic regression is applied to the top hidden layer. We use two layers of RNN of to match the model complexity of `RETAIN`.
- **RNN+$\alpha_M$**: One layer single directional RNN (hidden layer size 256) along time to generate the input embeddings $\mathbf{v}_1, \ldots, \mathbf{v}_i$. We use the MLP with a single hidden layer of size 256 to generate the visit-level attentions $\alpha_1, \ldots, \alpha_i$. We use the input embeddings $\mathbf{v}_1, \ldots, \mathbf{v}_i$ as the input to the MLP. This baseline corresponds to Figure 1a.
- **RNN+$\alpha_R$**: This is similar to **RNN+$\alpha_M$** but use the reverse-order RNN (hidden layer size 256) to generate the visit-level attentions $\alpha_1, \ldots, \alpha_i$. We use this baseline to confirm the effectiveness of generating the attentions using reverse time order.

The comparative visualization of the baselines are provided in Appendix D. We use the same implementation and training method for the baselines as described above. The details on the hyper-parameters, regularization and drop-out strategies for the baselines are described in Appendix B.

**Evaluation measures:** Model accuracy was measured by:

- **Negative log-likelihood** that measures the model loss on the test set. The loss can be calculated by Eq (1).
- **Area Under the ROC Curve (AUC)** of comparing $\widehat{y}_i$ with the true label $y_i$. AUC is more robust to imbalanced positive/negative prediction labels, making it appropriate for evaluation of classification accuracy in the heart failure prediction task.

We also report the bootstrap (10,000 runs) estimate of the standard deviation of the evaluation measures.

## 4.2 Heart Failure Prediction

**Objective:** Given a visit sequence $\mathbf{x}_1, \ldots, \mathbf{x}_T$, we predicted if a primary care patient will be diagnosed with heart failure (HF). This is a special case of DPM with a single disease outcome at the end of the sequence. Since this is a binary prediction task, we use the logistic sigmoid function instead of the Softmax in Step 5.

**Cohort construction:** From the source dataset, 3,884 cases are selected and approximately 10 controls are selected for each case (28,903 controls). The case/control selection criteria are fully described in the supplementary section. Cases have index dates to denote the date they are diagnosed with HF. Controls have the same index dates as their corresponding cases. We extract diagnosis codes, medication codes and procedure codes in the 18-months window before the index date.

**Training details:** The patient cohort was divided into the training, validation and test sets in a 0.75:0.1:0.15 ratio. The validation set was used to determine the values of the hyper-parameters. See Appendix B for details of hyper-parameter tuning.

Table 2: Heart failure prediction performance of `RETAIN` and the baselines

| Model | Test Neg Log Likelihood | AUC | Train Time / epoch | Test Time |
|---|---|---|---|---|
| LR | $0.3269 \pm 0.0105$ | $0.7900 \pm 0.0111$ | 0.15s | 0.11s |
| MLP | $0.2959 \pm 0.0083$ | $0.8256 \pm 0.0096$ | 0.25s | 0.11s |
| RNN | $0.2577 \pm 0.0082$ | $0.8706 \pm 0.0080$ | 10.3s | 0.57s |
| RNN+$\alpha_M$ | $0.2691 \pm 0.0082$ | $0.8624 \pm 0.0079$ | 6.7s | 0.48s |
| RNN+$\alpha_R$ | $0.2605 \pm 0.0088$ | $\mathbf{0.8717} \pm 0.0080$ | 10.4s | 0.62s |
| `RETAIN` | $\mathbf{0.2562} \pm 0.0083$ | $0.8705 \pm 0.0081$ | 10.8s | 0.63s |

**Results:** Logistic regression and MLP underperformed compared to the four temporal learning algorithms (Table 2). `RETAIN` is comparable to the other RNN variants in terms of prediction performance while offering the interpretation benefit.

Note that RNN+$\alpha_R$ model are a degenerated version of `RETAIN` with only scalar attention, which is still a competitive model as shown in table 2. This confirms the efficiency of generating attention weights using the RNN. However, RNN+$\alpha_R$ model only provides scalar visit-level attention, which is not sufficient for healthcare applications. Patients often receives several medical codes at a single visit, and it will be important to distinguish their relative importance to the target. We show such a case study in section 4.3.

Table 2 also shows the scalability of `RETAIN`, as its training time (the number of seconds to train the model over the entire training set once) is comparable to RNN. The test time is the number of seconds to generate the prediction output for the entire test set. We use the mini-batch of 100 patients when assessing both training and test times. RNN takes longer than RNN+$\alpha_M$ because of its two-layer structure, whereas RNN+$\alpha_M$ uses a single layer RNN. The models that use two RNNs (RNN, RNN+$\alpha_R$, `RETAIN`)[2] take similar time to train for one epoch. However, each model required a different number of epochs to converge. RNN typically takes approximately 10 epochs, RNN+$\alpha_M$ and RNN+$\alpha_R$ 15 epochs and `RETAIN` 30 epochs. Lastly, training the attention models (RNN+$\alpha_M$, RNN+$\alpha_R$ and `RETAIN`) for DPM would take considerably longer than L2D, because DPM modeling generates context vectors at each time step. RNN, on the other hand, does not require additional computation other than embedding the visit to its hidden layer to predict target labels at each time step. Therefore, in DPM, the training time of the attention models will increase linearly in relation to the length of the input sequence.

### 4.3 Model Interpretation for Heart Failure Prediction

We evaluated the interpretability of `RETAIN` in the HF prediction task by choosing a HF patient from the test set and calculating the contribution of the variables (medical codes in this case) to diagnostic prediction. The patient suffered from skin problems, *skin disorder* (SD), *benign neoplasm* (BN), *excision of skin lesion* (ESL), for some time before showing symptoms of HF, *cardiac dysrhythmia* (CD), *heart valve disease* (HVD) and *coronary atherosclerosis* (CA), and then a diagnosis of HF (Figure 3). We can see that skin-related codes from the earlier visits made little contribution to HF prediction as expected. `RETAIN` properly puts much attention to the HF-related codes that occurred in recent visits.

To confirm `RETAIN`'s ability to exploit the sequence information of the EHR data, we reverse the visit sequence of Figure 3a and feed it to `RETAIN`. Figure 3b shows the contribution of the medical codes of the reversed visit record. HF-related codes in the past are still making positive contributions, but not as much as they did in Figure 3a. Figure 3b also emphasizes `RETAIN`'s superiority to interpretable, but stationary models such as logistic regression. Stationary models often aggregate past information and remove the temporality from the input data, which can mistakenly lead to the same risk prediction for Figure 3a and 3b. `RETAIN`, however, can correctly digest the sequence information and calculates the HF risk score of 9.0%, which is significantly lower than that of Figure 3a.

Figure 3c shows how the contributions of codes change when selected medication data are used in the model. We added two medications from day 219: *antiarrhythmics* (AA) and *anticoagulants* (AC), both of which are used to treat *cardiac dysrhythmia* (CD). The two medications make a negative contributions, especially towards the end of the record. The medications decreased the positive contributions of *heart valve disease* and *cardiac dysrhythmia* in the last visit. Indeed, the HF risk

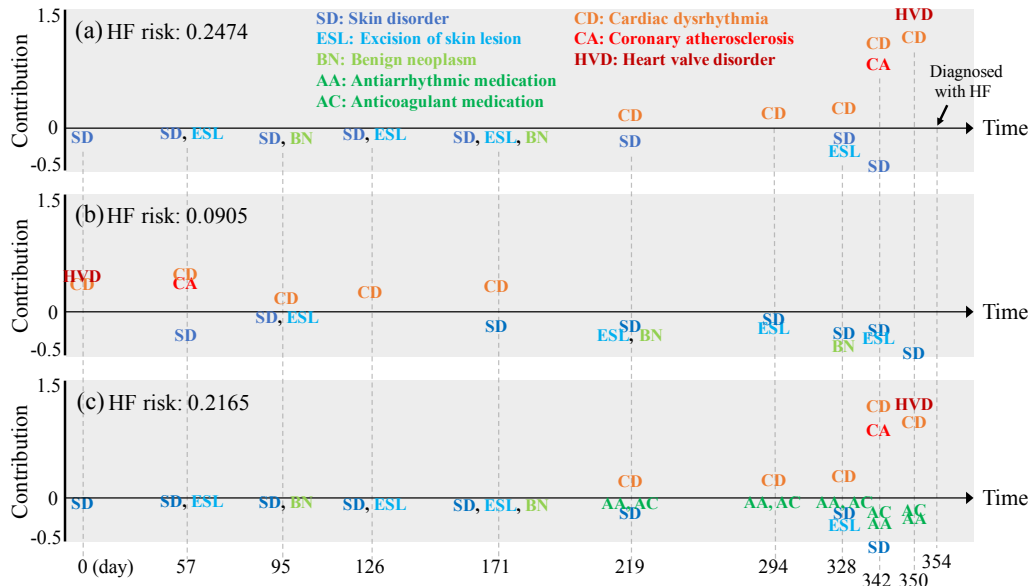

Figure 3: (a) Temporal visualization of a patient's visit records where the contribution of variables for diagnosis of heart failure (HF) is summarized along the $x$-axis (*i.e.* time) with the $y$-axis indicating the magnitude of visit and code specific contributions to HF diagnosis. (b) We reverse the order of the visit sequence to see if `RETAIN` can properly take into account the modified sequence information. (c) Medication codes are added to the visit record to see how it changes the behavior of `RETAIN`.

prediction (0.2165) of Figure 3c is lower than that of Figure 3a (0.2474). This suggests that taking proper medications can help the patient in reducing their HF risk.

## 5 Conclusion

Our approach to modeling event sequences as predictors of HF diagnosis suggest that complex models can offer both superior predictive accuracy and more precise interpretability. Given the power of RNNs for analyzing sequential data, we proposed `RETAIN`, which preserves RNN's predictive power while allowing a higher degree of interpretation. The key idea of `RETAIN` is to improve the prediction accuracy through a sophisticated attention generation process, while keeping the representation learning part simple for interpretation, making the entire algorithm accurate and interpretable. `RETAIN` trains two RNN in a reverse time order to efficiently generate the appropriate attention variables. For future work, we plan to develop an interactive visualization system for `RETAIN` and evaluating `RETAIN` in other healthcare applications.

## Footnotes

[1]For example, feeding visit embeddings in the original order to $\text{RNN}_\alpha$ and $\text{RNN}_\beta$ will generate the same $e_1$ and $\boldsymbol{\beta}_1$ for every time step $i = 1, 2, \ldots, T$. Moreover, in many cases, a patient's recent visit records deserve more attention than the old records. Then we need to have $e_{j+1} > e_j$ which makes the process computationally unstable for long sequences.

[2]The RNN baseline uses two layers of RNN, RNN+$\alpha_R$ uses one for visit embedding and one for generating $\alpha$, `RETAIN` uses each for generating $\alpha$ and $\boldsymbol{\beta}$

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
