[Supplementary Material]

# A  A method to use the timestamps

As before, we use $t_i^{(n)}$ to represent the timestamp of the $i$-th visit of the $n$-th patient. In the following, we suppress the superscript $(n)$ to avoid cluttered notation. Note that the timestamp $t_i$ can be anything that provides the temporal information of the $i$-th visit: the number of days from the first visit, the number of days between two consecutive visits, or the number of days until the index date of some event such as heart failure diagnosis.

In order to use the timestamps, we modify Step 2 and Step 3 in Section 2.2 as follows:

$$\mathbf{g}_i, \mathbf{g}_{i-1}, \ldots, \mathbf{g}_1 = \mathrm{RNN}_\alpha(\mathbf{v}_i^{'}, \mathbf{v}_{i-1}^{'}, \ldots, \mathbf{v}_1^{'}),$$
$$e_j = \mathbf{w}_\alpha^\top \mathbf{g}_j + b_\alpha, \quad \text{for} \quad j = 1, \ldots, i$$
$$\alpha_1, \alpha_2, \ldots, \alpha_i = \mathrm{Softmax}(e_1, e_2, \ldots, e_i)$$
$$\mathbf{h}_i, \mathbf{h}_{i-1}, \ldots, \mathbf{h}_1 = \mathrm{RNN}_\beta(\mathbf{v}_i^{'}, \mathbf{v}_{i-1}^{'}, \ldots, \mathbf{v}_1^{'})$$
$$\boldsymbol{\beta}_j = \tanh\left(\mathbf{W}_\beta \mathbf{h}_j + \mathbf{b}_\beta\right) \quad \text{for} \quad j = 1, \ldots, i,$$
$$\text{where } \mathbf{v}_i^{'} = [\mathbf{v}_i, t_i]$$

where we use $\mathbf{v}_i^{'}$, the concatenation of the visit embedding $\mathbf{v}_i$ and the timestamp $t_i$, to generate the attentions $\alpha$ and $\beta$. However, when obtaining the context vector $\mathbf{c}_i$ as per Step 4, we use $\mathbf{v}_i$, not $\mathbf{v}_i^{'}$ to match the dimensionality. The entire process could be understood such that we use the temporal information not to embed each visit, but to calculate the attentions for the entire visit sequence. This is consistent with our modeling approach where we lose the sequential information in embedding the visit with MLP, then recover the sequential information by generating the attentions using the RNN. By using the temporal information, specifically the log of the number of days from the first visit, we were able to improve the heart failure prediction AUC by 0.003 without any hyper-parameter tuning.

# B  Details of the experiment settings

## B.1  Hyper-parameter Tuning

We used the validation set to tune the hyper-parameters: visit embedding size $m$, $\mathrm{RNN}_\alpha$'s hidden layer size $p$, $\mathrm{RNN}_\beta$'s hidden layer size $q$, $L_2$ regularization coefficient, and drop-out rates.

$L_2$ regularization was applied to all weights except the ones in $\mathrm{RNN}_\alpha$ and $\mathrm{RNN}_\beta$. Two separate drop-outs were used on the visit embedding $\mathbf{v}_i$ and the context vector $\mathbf{c}_i$. We performed the random search with predefined ranges $m, p, q \in \{32, 64, 128, 200, 256\}$, $L_2 \in \{0.1, 0.01, 0.001, 0.0001\}$, $dropout_{\mathbf{v}_i}, dropout_{\mathbf{c}_i} \in \{0.0, 0.2, 0.4, 0.6, 0.8\}$. We also performed the random search with $m$, $p$ and $q$ fixed to 256.

The final value we used to train `RETAIN` for heart failure prediction is $m, p, q = 128$, $dropout_{\mathbf{v}_i} = 0.6$, $dropout_{\mathbf{c}_i} = 0.6$ and 0.0001 for the $L_2$ regularization coefficient.

## B.2  Code Grouper

Diagnosis codes, medication codes and procedure codes in the dataset are respectively using International Classification of Diseases (ICD-9), Generic Product Identifier (GPI) and Current Procedural Terminology (CPT).

Diagnosis codes are grouped by Clinical Classifications Software for ICD-9[16] which reduces the number of diagnosis code from approximately 14,000 to 283. Medication codes are grouped by Generic Product Identifier Drug Group[24] which reduces the dimension to from approximately 151,000 to 96. Procedure codes are grouped by Clinical Classifications Software for CPT[17], which reduces the number of CPT codes from approximately 9,000 to 238.

## B.3  Training Specifics of the Basline Models

- **LR**: We use 0.01 $L_2$ regularization coefficient for the logistic regression weight.
- **MLP**: We use drop-out rate 0.6 on the output of the hidden layer. We use 0.0001 $L_2$ regularization coefficient for the hidden layer weight and the logistic regression weight.

Table 3: Qualifying ICD-9 codes for heart failure

| ICD-9 Code | Description |
|---|---|
| 398.91 | Rheumatic heart failure (congestive) |
| 402.01 | Malignant hypertensive heart disease with heart failure |
| 402.11 | Benign hypertensive heart disease with heart failure |
| 402.91 | Unspecified hypertensive heart disease with heart failure |
| 404.01 | Hypertensive heart and chronic kidney disease, malignant, with heart failure and with chronic kidney disease stage I through stage IV, or unspecified |
| 404.03 | Hypertensive heart and chronic kidney disease, malignant, with heart failure and with chronic kidney disease stage V or end stage renal disease |
| 404.11 | Hypertensive heart and chronic kidney disease, benign, with heart failure and with chronic kidney disease stage I through stage IV, or unspecified |
| 404.13 | Hypertensive heart and chronic kidney disease, benign, with heart failure and chronic kidney disease stage V or end stage renal disease |
| 404.91 | Hypertensive heart and chronic kidney disease, unspecified, with heart failure and with chronic kidney disease stage I through stage IV, or unspecified |
| 404.93 | Hypertensive heart and chronic kidney disease, unspecified, with heart failure and chronic kidney disease stage V or end stage renal disease |
| 428.0 | Congestive heart failure, unspecified |
| 428.1 | Left heart failure |
| 428.20 | Systolic heart failure, unspecified |
| 428.21 | Acute systolic heart failure |
| 428.22 | Chronic systolic heart failure |
| 428.23 | Acute on chronic systolic heart failure |
| 428.30 | Diastolic heart failure, unspecified |
| 428.31 | Acute diastolic heart failure |
| 428.32 | Chronic diastolic heart failure |
| 428.33 | Acute on chronic diastolic heart failure |
| 428.40 | Combined systolic and diastolic heart failure, unspecified |
| 428.41 | Acute combined systolic and diastolic heart failure |
| 428.42 | Chronic combined systolic and diastolic heart failure |
| 428.43 | Acute on chronic combined systolic and diastolic heart failure |
| 428.9 | Heart failure, unspecified |

- **RNN**: We use drop-out rate 0.6 on the outputs of both hidden layers. We use 0.0001 $L_2$ regularization coefficient for the logistic regression weight. The dimension size of both hidden layers is 256.

- **RNN+$\alpha_M$**: We use drop-out rate 0.4 on the output of the hidden layer and 0.6 on the output of the context vector $\sum_i \alpha_i \mathbf{v}_i$. We use 0.0001 $L_2$ regularization coefficient for the hidden layer weight of the MLP that generates $\alpha$'s and the logistic regression weight. The dimension size of the hidden layers in both RNN and MLP is 256.

- **RNN+$\alpha_R$**: We use drop-out rate 0.4 on the output of the hidden layer and 0.6 on the output of the context vector $\sum_i \alpha_i \mathbf{v}_i$. We use 0.0001 $L_2$ regularization coefficient for the hidden layer weight of the RNN that generates $\alpha$'s and the logistic regression weight. The dimension size of the hidden layers in both RNNs is 256.

## B.4    Heart Failure Case/Control Selection Criteria

Case patients were 40 to 85 years of age at the time of HF diagnosis. HF diagnosis (HFDx) is defined as: 1) Qualifying ICD-9 codes for HF appeared in the encounter records or medication orders. Qualifying ICD-9 codes are displayed in Table 3. 2) a minimum of three clinical encounters with qualifying ICD-9 codes had to occur within 12 months of each other, where the date of diagnosis was assigned to the earliest of the three dates. If the time span between the first and second appearances of the HF diagnostic code was greater than 12 months, the date of the second encounter was used as the first qualifying encounter. The date at which HF diagnosis was given to the case is denoted as HFDx. Up to ten eligible controls (in terms of sex, age, location) were selected for each case, yielding an overall ratio of 9 controls per case. Each control was also assigned an index date, which is the HFDx of the matched case. Controls are selected such that they did not meet the operational criteria for HF diagnosis prior to the HFDx plus 182 days of their corresponding case. Control subjects were

required to have their first office encounter within one year of the matching HF case patient's first office visit, and have at least one office encounter 30 days before or any time after the case's HF diagnosis date to ensure similar duration of observations among cases and controls.

## C Results on disease progression modeling

**Objective:** Given a sequence of visits $\mathbf{x}_1, \ldots, \mathbf{x}_T$, the goal of DPM is, for each time step $i$, to predict the codes occurring at the next visit $\mathbf{x}_2, \ldots, \mathbf{x}_{T+1}$. However, as we are interested in the disease progression, we create a separate set of labels $\mathbf{y}_1, \ldots, \mathbf{y}_T$ that do not contain non-diagnosis codes such as medication codes or procedure codes. Therefore $\mathbf{y}_i$ will contain diagnosis codes from the next visit $\mathbf{x}_{i+1}$.

**Dataset:** We divide the entire dataset described in Table 1 into 0.75:0.10:0.15 ratio, respectively for training set, validation set, and test set.

**Baseline:** We use the same baseline models we used for HF prediction. However, since we are predicting 283 binary labels now, we replace the logistic regression function with the Softmax function. The drop-out and $L_2$ regularization policies remain the same.

For LR and MLP, at each step $i$, we aggregate maximum ten past input vectors[3] $\mathbf{x}_{i-9}, \ldots, \mathbf{x}_i$ to create a pseudo-context vector $\widehat{\mathbf{c}}_i$. LR applies the Softmax function on top of $\widehat{\mathbf{c}}_i$. MLP places a hidden layer on top of $\widehat{\mathbf{c}}_i$ then applies the Softmax function.

**Evaluation metric:** We use the negative log likelihood Eq (1) on the test set to evaluate the model performance. We also use Recall@$k$ as an additional metric to measure the prediction accuracy.

- **Recall@$k$:** Given a sequence of visits $\mathbf{x}_1, \ldots, \mathbf{x}_T$, we evaluate the model performance based on how accurately it can predict the diagnosis codes $\mathbf{y}_1, \ldots, \mathbf{y}_T$. We use the average Recall@$k$, which is expressed as below,

$$\frac{1}{N} \sum_{n=1}^{N} \frac{1}{T^{(n)}} \sum_{i=1}^{T^{(n)}} \text{Recall@}k(\widehat{\mathbf{y}}_i), \quad where \quad \text{Recall@}k(\widehat{\mathbf{y}}_i) = \frac{|\text{argsort}(\widehat{\mathbf{y}}_i)[:k] \cap nonzero(\mathbf{y}_i)|}{|nonzero(\mathbf{y}_i)|}$$

where $argsort$ returns a list of indices that will decrementally sort a given vector and $nonzero$ returns a list of indices of the coordinates with non-zero values. We use Recall@$k$ because of its similar nature to the way a human physician performs the differential diagnostic procedure, which is to generate a list of most likely diseases for an undiagnosed patient, then perform medical practice until the true disease, or diseases are determined.

**Prediction accuracy:** Table 4 displays the prediction performance of `RETAIN` and the baselines. We use $k = 5, 10$ for Recall@$k$ to allow a reasonable number of prediction trials, as well as cover complex patients who often receive multiple diagnosis codes at a single visit.

RNN shows the best prediction accuracy for DPM. However, considering the purpose of DPM, which is to assist doctors to provide quality care for the patient, black-box behavior of RNN makes it unattractive as a clinical tool. On the other hand, `RETAIN` performs as well as other attention models, only slightly inferior to RNN, provides full interpretation of its prediction behavior, making it a feasible solution for clinical applications.

The interesting finding in Table 4 is that MLP is able to perform as accurately as RNN+$\alpha_M$ in terms of Recall@10. Considering the fact that MLP uses aggregated information of past ten visits, we can assume that DPM depends more on the frequency of disease occurrences rather than the order in which they occurred. This is quite different from the HF prediction task, where stationary models (LF, MLP) performed significantly worse than sequential models.

## D Illustration and comparison of the baselines

Figure 4 illustrates the baselines used in the experiments and shows the relationship among them.

Table 4: Disease progression modeling performance of `RETAIN` and the baselines

| Model | Negative Likelihood | Recall@5 | Recall@10 |
|---|---|---|---|
| LR | 0.0288 | 43.15 | 55.84 |
| MLP | 0.0267 | 50.72 | 65.02 |
| RNN | **0.0258** | **55.18** | **69.09** |
| RNN+$\alpha_M$ | 0.0262 | 52.15 | 65.81 |
| RNN+$\alpha_R$ | 0.0259 | 53.89 | 67.45 |
| `RETAIN` | 0.0259 | 54.25 | 67.74 |

(a)       (b)       (c)       (d)       (e)

Figure 4: Graphical illustration of the baselines: (a) Logistic regression (LR), (b) Multilayer Perceptron (MLP), (c) Recurrent neural network (RNN), (d) RNN with attention vectors generated via an MLP (RNN+$\alpha_M$), (e) RNN with attention vectors generated via an RNN (RNN+$\alpha_R$). `RETAIN` is given in Figure 1b.

## Footnotes

[3]We also tried aggregating all past input vectors $\mathbf{x}_1, \ldots, \mathbf{x}_i$, but the performance was slightly worse than using just ten.