[Reviews · NeurIPS 2016]

Reviewer 1

Summary

Summary: Authors propose an attention based neural network model for learning from longitudinal patient visit data. The proposed approach weights both the visit-level and variable-level attention, and attention vectors in reverse order. The authors claim the model is interpretable, since they’re able to extract a contribution coefficient for each variable.

Qualitative Assessment

Overall: Interpretable models are important (especially in healthcare), and the proposed method makes sense, but I have a few concerns/questions regarding the work: *“Accuracy and interpretation are two goals of any successful predictive model.” This is a strong statement, made both in the abstract and introduction. I suspect there’s a subset of the ML community that might disagree with this statement, i.e., a subset the only cares about accuracy and where interpretability is not important. *In the Heart Failure Prediction problem, the authors make the fundamental mistake of defining the prediction task in terms of the outcome. I.e., the input data are extracted in the 18-month window before the index date. Given a new test patient, this same preprocessing would require knowing not only the diagnosis of the patient (i.e., the label) but also when the diagnosis occurred. Such a setup is often used to measure the correlation between input and output, but should not be used for testing predictive models, since such a model is of little clinical utility and can be strongly affected by how the index event is defined for “negative” patients. Instead the authors should make a prediction for each visit. This varies how far one is from the index event, and doesn't rely on label information for pre-processing. *In the evaluation, the authors compare their proposed approach to several baselines. Including logistic regression. As the authors state, the proposed method outperforms the other methods because the baseline methods ignore the temporal aspects of the problem. In some sense, this isn’t really a fair comparison, since there are standard ways of incorporating temporality into a feature vector for input into LR or an SVM. Would be good to see such comparisons. *In terms of interpretability, the contributions are for individual variables, but often in healthcare we see more complex interactions between variables. How can the model identify such important interactions?

Confidence in this Review

3-Expert (read the paper in detail, know the area, quite certain of my opinion)


Reviewer 2

Summary

This paper aims to provide more interpretable models from clinical time series data by relying more on recent time points than distant ones for making predictions. Predictive accuracy is compared to the current state of the art on a set of clinical data.

Qualitative Assessment

The problem addressed (model interpretability) is an important one, and certainly a model that a clinician can understand would be preferable to a black-box model with similar accuracy. The assumption that focusing on recent timepoints may help create such models is reasonable on the face of it, but is not well justified in the paper (theoretically or empirically), and it is unclear whether the approach would generalize to any other problems. The lack of confidence in the results and lack of clear application to any other problem limited my enthusiasm. Fundamentally, the paper demonstrates that their approach has similar predictive accuracy to the state of the art on one dataset, but fails to demonstrate that it has any advantages that make it superior despite the similar accuracy. Key questions that need to be addressed: 1) Handling of irregularly sampled time series: It seems the authors have to assume the same sampling rate for all patients (as on pg 2 they refer to the i and i+1th visit as a time step). However, visits are irregularly spaced and will certainly occur at different frequencies for different patients. How does this affect the algorithm? Similarly, there may be a cluster of frequent visits during an acute illness, but these will likely have redundant information and should be collapsed into a single episode. 2) Why is the assumption of more recent events being most important justified? The entire paper rests on this assumption, but it is not obvious that this should be the case. For example, duration of diabetes is often an important variable for predicting complications, but there we need to understand the onset, not the most recent mention of diabetes in a medical record. 3) A lot of detail on the experiments is missing. It's not clear where the data come from or how this definition of heart failure was created or validated. Further many decisions were made in assigning diagnosis dates (and dates for controls) that are never justified. 4) The evaluation of interpretability contains only anecdotes related to the single experiment. Were any tests done to determine if the proposed approach leads to predictions that are more interpretable by humans? I would think that should be a key evaluation for this paper. Most concerning, though, is that it's stated that "RETAIN properly puts much attention to the HF-related codes that occurred in recent visits." According to the method for finding HF cases, it seems that in most cases the HF codes should be at or after the dx-date. It seems that the time series was not appropriately segmented, so that data after diagnosis is being used for the prediction task. Finding that HF codes predict heart failure is not a clinically meaningful model. 5) How well does this method do on other datasets? In particular, are there any experiments on data where ground truth is known? EHRs are very noisy and heart failure diagnosis can be ambiguous in these data. The use of only a single experiment in this case is less convincing.

Confidence in this Review

2-Confident (read it all; understood it all reasonably well)


Reviewer 3

Summary

The paper proposes Reverse Time Attention (RETAIN) model for analyzing time series, which achieve high accuracy on EHR data, while retaining clinical interpretability. It extends Recurrent Neural Network in two key aspects (1) the model analyzes data in reverse order inspired by physicians dianogsis, starting from the most recent time stamp, and (2) has an addtional layer to explicitly analyze the influence of individual variables at each time step on the final prediction. The 2nd aspect helps interpret the prediction process, which makes sense in medical analysis. The results on a EHR dataset show the proposed method obtain comparable accuracy and speed to other methods, while have advantage of process interpretability.

Qualitative Assessment

Overall the paper has good quality, and it uses a nice combination of sophisticated techniques to achieve good results on interesting medical problem domains. In terms of clarity, the paper is reasonably clear and well written. This work has moderate originality that uses recent advances from attention-based models with the goal of improving the interpretability of the RNN models. However, I was left with some questions (see below) about details of the model and the results which hopefully the authors can address in their feedback. - In equations (step 2) and (step 3), the authors use different non-linear activation functions for two RNNs ( they use "softmax" for \alpha-RNN and "tanh" for \beta-RNN). Why the authors did not use the same activation functions for both RNNs? - In equation (step4), the authors multiply the \alpha nad \beta parameters to generate the context vectors. This element-wise multiplication may cause identifiability issues (for example, multiplying \alpha and dividing \beta by the same constant does not change the corresponding context vector) for learning \alpha and \beta. It would be interesting if the authors explain how they have addressed this issue. - In terms of accuracy, The performance of the proposed method is not significantly better than the simple RNN models. Therefore, the contribution of the paper is in that it can find influential factors in the input data. However, Fig. 3 is insufficient to support the authors claim. First, only one person is involved in the experiments. Second, there is no comparison with other interpretable models such as Logistic Regression model. It would be interesting to see how much interpretable the proposed model is in comparison to other simple models.

Confidence in this Review

2-Confident (read it all; understood it all reasonably well)


Reviewer 4

Summary

This paper tackles the problem of accurate and interpretable predictions of healthcare issues. The authors propose two-level neural attention model with RNN, one for temporal visits and another for medical codes(dimensions in each visit). They conduct two experiments: learning to diagnose (LTD) and disease progression modeling (DPM) on Electronic Health Record (EHR) data, and show that the performances are comparative and also results are interpretable.

Qualitative Assessment

Interpretability is an important issue and the idea in this paper is interesting, and, to the best of my knowledge, seems novel. By using two-level neural attention model with RNN, it can achieve the comparable performance, and also shows interpretable results, both in temporally and medical codes, which is nice. The experiments look to validate the proposed approach, even though RNNs are still slightly better as one can expect. The paper is clearly written. More comments: - In table 2, RNN+\alpha_M took shorter time than RNN, but RNN+\alpha_R, which has the similar structure, also took longer. Do you have any ideas why? - In figure 3(a), ESL and SD started to have more negative contributions in more recent visits. Do you have any intuitions why?

Confidence in this Review

2-Confident (read it all; understood it all reasonably well)


Reviewer 5

Summary

This paper presents an innovative attention model to make predictive model interpretable using deep learning approaches in EHRs. This is an important step in healthcare applications. The performance is state of the art (AUC 0.8717 for heart failure prediction).

Qualitative Assessment

This paper presents a clever use of attention model to make RNNs interpretable. However, the attention model is simple; the goal is to keep the representation learning part simple and interpretable. Can author provide insight why this representation is comparable to regular RNNs? In the authors' experience, would a "deeper" architecture achieves a better performance in EHRs?

Confidence in this Review

2-Confident (read it all; understood it all reasonably well)